# Sleep and Executive Functioning in Pediatric Traumatic Brain Injury Survivors after Critical Care

**DOI:** 10.3390/children9050748

**Published:** 2022-05-19

**Authors:** Cydni N. Williams, Cindy T. McEvoy, Miranda M. Lim, Steven A. Shea, Vivek Kumar, Divya Nagarajan, Kurt Drury, Natalia Rich-Wimmer, Trevor A. Hall

**Affiliations:** 1Pediatric Critical Care and Neurotrauma Recovery Program, Oregon Health & Science University, Portland, OR 97239, USA; kumarv@ohsu.edu (V.K.); richwimn@ohsu.edu (N.R.-W.); halltr@ohsu.edu (T.A.H.); 2Division of Pediatric Critical Care, Department of Pediatrics, Oregon Health & Science University, Portland, OR 97239, USA; nagarajd@ohsu.edu (D.N.); druryk@ohsu.edu (K.D.); 3Division of Neonatology, Department of Pediatrics, Oregon Health & Science University, Portland, OR 97239, USA; mcevoyc@ohsu.edu; 4Department of Neurology, Oregon Health & Science University, Portland, OR 97239, USA; lmir@ohsu.edu; 5Division of Pulmonary and Critical Care Medicine, Department of Medicine, Oregon Health & Science University, Portland, OR 97239, USA; 6Department of Behavioral Neuroscience, Oregon Health & Science University, Portland, OR 97239, USA; 7VA Portland Health Care System, Oregon Health & Science University, Portland, OR 97239, USA; 8Oregon Institute of Occupational Health Sciences, Oregon Health & Science University, Portland, OR 97239, USA; sheast@ohsu.edu; 9Division of Pediatric Psychology, Department of Pediatrics, Oregon Health & Science University, Portland, OR 97239, USA

**Keywords:** brain injury, sleep, pediatric, critical care

## Abstract

Over 50,000 children are hospitalized annually for traumatic brain injury (TBI) and face long-term cognitive morbidity. Over 50% develop sleep/wake disturbances (SWDs) that can affect brain development and healing. We hypothesized SWDs would portend worse executive function outcomes in children aged 3–18 years with TBI 1–3 months after hospital discharge. SWDs were defined using the Sleep Disturbances Scale for Children (t-scores ≥ 60). Outcomes included the Global Executive Composite (GEC, t-score) from the Behavior Rating Inventory of Executive Function, Second and Preschool Editions, and multiple objective executive function assessments combined through Principal Components Analysis into a Neurocognitive Index (NCI, z-score). Multiple linear regression evaluated associations between SWDs and executive function outcomes, controlling for covariates. Among 131 children, 68% had clinically significant SWDs, which were associated with significantly worse median scores on the GEC (56 vs. 45) and NCI (−0.02 vs. 0.42; both *p* < 0.05). When controlling for baseline characteristics and injury severity in multivariable analyses, SWDs were associated with worse GEC (β-coefficient = 7.8; 95% Confidence Interval = 2.5, 13.1), and worse NCI (β-coefficient = −0.4; 95% Confidence Interval = −0.8, −0.04). SWDs in children with TBI are associated with worse executive function outcomes after hospital discharge, and may serve as modifiable targets to improve outcomes.

## 1. Introduction

Sleep wake disturbances (SWDs) affect over 50% of the 50,000 children surviving critical care hospitalization for traumatic brain injury (TBI) each year in the United States [1,2,3,4,5]. Despite the documented benefits of healthy and restorative sleep on brain development and healing [6,7,8,9,10,11], little is known about how SWDs impact cognitive recovery during the acute recovery phase after pediatric TBI, especially among children hospitalized in the Pediatric Intensive Care Unit (PICU) for mild, moderate, or severe TBI.

Previous research has shown that 52% of youth with a TBI presenting to a PICU follow-up clinic one to three months post-hospital discharge have SWDs, with 84% of them exhibiting phenotypes consistent with insomnia and circadian disturbance, though multiple SWD phenotypes often coexist [5]. Prior findings also show TBI results in an increase in SWDs compared to healthy cohorts or orthopedic injury controls [12,13]. This is important because healthy sleep is vital for brain maturation and memory consolidation, which impact cognitive processes such as executive functioning important for normal childhood development and academic achievement [6,7,8,9,10,11,12,14,15,16,17,18,19]. Disturbances in executive function are associated with a number of problematic long-term outcomes for pediatric TBI survivors such as poor adaptive skills, personality changes, impaired social relationships, behavioral problems, and worse academic achievement [20,21,22,23,24,25,26].

With issues related to executive dysfunction being present during the acute recovery phase and persisting >10 years after injury in children across all ages and the spectrum of TBI severity [20,27,28,29,30], it is of paramount importance to empirically explore the relationship of SWDs and executive functioning, as sleep is a potentially modifiable target for intervention. Few studies, however, have evaluated SWDs as a risk factor for cognitive dysfunction after pediatric TBI despite the known benefits of healthy sleep on cognition and links between SWDs and cognitive performance in other non-TBI diseases [18,19,30].

The primary aim of this study was to evaluate the relationship between parent reported outcomes related to sleep and executive functioning in a cohort of pediatric TBI survivors during the acute recovery phase after critical care. A secondary aim was to explore the relationship between parent-reported sleep outcomes and direct clinical assessment of executive functioning, utilizing neuropsychological evaluations. Given the importance of sleep to brain development, overall health, and repair after injury, we hypothesized that SWDs, as reported by parents, would be associated with poorer executive functioning outcomes on both subjective proxy ratings and direct assessment.

## 2. Materials and Methods

### 2.1. Participants and Procedures

We performed a secondary cross-sectional analysis of data collected during a prospective observational study of children aged 3 to 18-years admitted for TBI surviving to hospital discharge and referred for follow-up in the Pediatric Critical Care and Neurotrauma Recovery Program (PCCNRP). Eligible patients (N = 167) completed a follow-up visit between one and three months after hospital discharge between September 2018 and November 2021. Consecutive patients presenting to the clinic were included if they had complete sleep measure data (N = 145) and complete results for at least one of the cognitive outcome measures (N = 131). Participants were excluded for missing or incomplete measures or age <3 years. The PCCNRP referrals, follow-up patterns, and program details have been previously described [31,32]. Briefly, the PCCNRP receives systematic referrals for all pediatric trauma patients admitted to our Level 1 pediatric trauma and tertiary care academic children’s hospital with follow-up rates consistently at >70%. The Institutional Review Board at Oregon Health and Science University approved the study procedures with a waiver of consent.

### 2.2. Demographic and Clinical Variables

Data were extracted from a longitudinal database maintained by the PCCNRP for all clinical visits. Traumatic brain injury severity was measured by admission Glasgow Coma Scale (GCS; pediatric version) [33], Injury Severity Score (ISS) [34], and Abbreviated Injury Scale (AIS; version 2015) scores [35,36]. Scores for ISS and AIS are assigned by trained trauma program staff for submission to the National Trauma Data Bank per published data standards [37,38]. We also approximated injury severity by collecting length of stay in the hospital and PICU, and by evaluating the need for any critical care interventions including mechanical ventilation, neurosurgical intervention, vasopressor infusion, central venous line placement, arterial line placement, intracranial pressure monitor, dialysis, therapy for refractory status epilepticus, extracorporeal support, and cardiopulmonary resuscitation (CPR). Intubation and mechanical ventilation were not recorded if only utilized during operations or procedures. Patient premorbid medical, psychiatric (e.g., anxiety, depression), or neurodevelopmental disorders (e.g., learning disability, attention-deficit/hyperactivity disorder, autism spectrum disorder) were documented from inpatient and clinic notes. Hospital outcomes were also recorded, including discharge to inpatient rehabilitation or home, and change from baseline Functional Status Scale (FSS) score [39]. FSS scores are assigned by PICU attending physicians to reflect pre-hospital, discharge, and follow-up status [40,41]. Diagnoses, active treatments, and recommendations were recorded from clinician follow-up notes.

### 2.3. Sleep Disturbance

The primary exposure was sleep disturbance defined by the Sleep Disturbance Scale for Children (SDSC) measured one to three months after hospitalization [42]. The SDSC is shown to be reliable and valid in PICU populations with acquired brain injuries [43]. The sleep disturbance group was defined by a t-score ≥60 on any of the SDSC domain scales consistent with prior work [5]. The SDSC is a 26-item standardized parent/caregiver proxy assessment of child and adolescent sleep behaviors for use in children ages 3 to 18 years. Parents report across six domains of sleep disturbances (Disorders of Initiating and Maintaining Sleep, Sleep Breathing Disorders, Disorders of Arousal, Sleep-Wake Transition Disorders, Disorders of Excessive Somnolence, and Sleep Hyperhidrosis) using a 5-point Likert scale with anchors “Never” to “Always (Daily).” Item responses are summed to calculate each factor score and converted to t-scores for ease of interpretation, with higher scores indicating more sleep disturbance. Domain t-scores ≥ 60 are reported to portend moderate or severe risk of clinically important sleep disorders across six groups of sleep disorders in children and adolescents [42]. Participants with ≥8 missing items on the SDSC were excluded for incompleteness as above. An additional seven forms had <8 missing items, and we imputed those missing items to a score of 1 (“Never”) for SDSC score calculations. The SDSC was only given in English during clinic visits, meaning language barriers contributed to missing SDSC form data.

### 2.4. Cognitive Assessments

The primary outcome utilized in this study was a combined Global Executive Composite (GEC) from the Behavior Rating Inventory of Executive Function, Second Edition (BRIEF-2, *n* = 88), and the Preschool Edition (BRIEF-P, *n* = 13). The scores were combined, as both represent overall function and there was no significant difference in distributions of the GEC between the two measures overall or by sleep disturbance group. The BRIEF-2 is recommended by the National Institutes of Health as a common data element for TBI research and has been validated across ages with good psychometric performance [44,45,46]. The BRIEF-P is a standardized 63-item parent/caregiver proxy measure of behaviors related to executive function in youth ≥3–5 years of age [47]. The BRIEF-2 is a standardized 63-item parent/caregiver proxy measure of behaviors related to executive function in youth >5–18 years of age [44]. Both BRIEF versions are meant to reflect estimates of daily life skills and performance related to executive function. Forms are scored according to the manual for handling missing data, including the exclusion of forms with >12 total missing, >1 missing for subscale scores, and imputation of single missing items on a subscale to ‘Never’. Raw scores are converted to t-scores standardized to patient age and sex for interpretation. The GEC is considered clinically elevated when ≥65 [44].

Objective cognitive assessments performed by a board-certified Pediatric Neuropsychologist within the executive function domain were evaluated as secondary outcomes in this study. Test utilization varied by age. Wide-Range Achievement Test, 4th Edition (WRAT-4), and 5th Edition (WRAT-5) are brief measures of academic functioning, and the Word Reading subtest was used as a proxy estimate for premorbid cognitive functioning [48,49]. Either the Children’s Memory Scale (CMS; ages 8–16 years) Numbers subtest [50], or the Wechsler Adult Intelligence Scale, 4th Edition (WAIS-IV; ages 17–18 years) Digit Span subtest [51] were administered to measure attention, rote memory, and working memory. The Child and Adolescent Memory Profile (ChAMP) Lists subtest [52] gauged verbal learning and general memory ability. Parts of the Verbal Fluency and Trail Making components of the Delis–Kaplan Executive Function System (D–KEFS) were utilized as measures of language, processing speed, and cognitive flexibility [53]. Either the Wechsler Intelligence Scale for Children, 5th Edition (WISC-V; ages 8–16 years) [54], or the WAIS-IV (ages 17–18 years) Coding and Symbol Search subtests were administered as processing speed measures. Results of these assessments yield scaled scores (ss; M = 10, SD = 3) or standard scores (SS; M = 100, SD = 15). Results are standardized to age and are considered clinically important when <7 or <85, respectively. The rationale for use of these assessments was reported previously [31,32,55,56]. Missing scores on subtests were related to complications such as orthopedic injury, vision impairment, poor behavioral cooperation precluding completion of some tasks, and the need for virtual only visits for some patients assessed at the beginning of the COVID-19 pandemic. Clinical interview with the patient and their parent/caregiver as part of the clinical assessment with a board-certified Pediatric Neuropsychologist was used to diagnose new psychological or emotional problems.

### 2.5. Data Analyses

We used chi-square with Fischer Exact correction for expected cell counts <10 and Mann–Whitney U tests to compare our analysis sample to those excluded with missing or incomplete data to assess for bias in our results (Appendix A). The excluded cohort had worse injury severity and were more likely Hispanic, which is consistent with the reasons for missing data noted above. Descriptive statistics were reported for demographic and clinical characteristics, as well as primary predictor and outcome variables. The majority of continuous variables were not normally distributed and reported as median with interquartile range (IQR).

In an effort to reduce type I error with multiple comparisons, we utilized Principle Components Analysis (PCA) to create a Neurocognitive Index (NCI) from the objective cognitive assessment scores listed above, which is consistent with prior work [32]. Details of the PCA can be found in Appendix A. The NCI represents a standardized z-score and ranged from −2.5 to 2.1 in our sample. Notably, only children with data for all components entered into the PCA are assigned an NCI (*n* = 79). The D–KEFS measures included in the NCI are only utilized in children ≥8 years of age, which excluded younger children from this secondary outcomes analysis.

We used chi-square and Mann–Whitney U tests as appropriate to compare variables between SWD groups (Table 1) in order to assess for confounders. We evaluated differences in outcomes (Table 2) between SWD groups using *t*-tests and Mann–Whitney U tests as appropriate. We used Spearman correlation (rs) to evaluate relationships between scores on the SDSC, injury severity measures, BRIEF, and objective neurocognitive assessments. We used simple linear regression to evaluate demographic and clinical variables in relation to both the GEC and the NCI in order to evaluate covariates for multiple regression models (Table 3). Results were reported as β-coefficients with 95% confidence interval (95% CI). Some continuous predictors were mathematically transformed prior to entering into regressions given non-normal distributions, and others were categorized based on quartiles if unable to be transformed into normal distributions. Multiple linear regression was used to evaluate the contribution of sleep disturbances to the GEC in our primary analysis, and the contribution of sleep disturbance to the NCI in our secondary exploratory analysis (Table 4 and Table 5). Based on prior studies [20,29,48,49,57,58,59], we planned a priori to include WRAT-4/5 word reading scores (estimate of baseline cognitive status) and injury severity in multivariable models. Other potential covariates and confounders for the full models were identified from bivariate analyses described above and entered into the model at a significance level of *p* < 0.1. Variables were tested individually for collinearity and multi-collinearity. No variables were excluded at this step as the variance inflation factors were all <5 and correlation coefficients <0.6. Backward stepwise regression was employed given the limited sample size to remove variables at the adjusted *p* > 0.1 level to produce final reduced models for each outcome analysis. Results of the full and reduced models are reported. IBM SPSS Version 27 was used for all statistical analyses, and significance was defined as *p* < 0.05.

## 3. Results

Among the 167 patients completing a follow-up visit, we evaluated 131 (78%) children aged 3 to 18 years a median of 49 days after critical care hospitalization for TBI (Table 1). Most were categorized as mild TBI severity (mild 43%, mild complicated 37%, moderate 9%, severe 11%). The majority of the participants were male (60%) and had Medicaid insurance (60%). Thirty-seven (28%) patients had one or more prior medical (9%), psychiatric (10%), or neurodevelopmental (18%) diagnoses. Half of the patients (*n* = 61, 47%) required at least one critical care intervention, including 30% requiring intubation and mechanical ventilation. Nine (7%) were discharged to inpatient rehabilitation facilities and 25 (19%) had a decline from pre-admission baseline in functional abilities measured by the FSS at the time of follow-up.

Overall, 68% (*n* = 89) of the sample had clinically relevant SWD (any t-score ≥ 60 on the SDSC). The internal consistency for the SDSC in our sample was excellent (α = 0.85). Disturbances in multiple sleep domains were found in 43% of children. Disturbances were found in all six sleep domains measured by the SDSC subscales: Disorders of Initiating and Maintaining Sleep (53%), Sleep Breathing Disorders (11%), Disorders of Arousal (15%), Sleep-Wake Transition Disorders (31%), Disorders of Excessive Somnolence (10%), and Sleep Hyperhidrosis (11%). Eleven (12%) patients with SWDs reported taking melatonin at the time of evaluation, and 38 (43%) were prescribed a new sleep treatment at the clinical evaluation. As seen in Table 1, SWDs were significantly associated with pre-admission neurodevelopmental comorbidities (*p =* 0.007). No significant difference was found for the development of new psychological diagnoses (e.g., depression, anxiety, stress disorders) between SWD groups. A significant association was also found between SWDs and some critical care interventions in that those with more severe injury evidenced by intracranial pressure monitors, arterial lines, and targeted temperature management reported less SWDs (all *p* < 0.05), but numbers were low for these interventions. Overall, no significant differences were found between TBI severity groups (defined by GCS score) and SDSC domain scores (Appendix A).

Table 2 shows average outcome scores by SWD group. Patients in the SWD group had significantly higher mean scores on the GEC, indicating worse function, compared to children without SWDs after TBI with nearly a full standard deviation higher score in the SWD group (56.8 versus 47.4; *p* < 0.001). Average values of NCI were not statistically different between SWD groups, but the distribution of scores was skewed within groups. Significantly different distributions of NCI scores were found in children with SWDs (median = −0.02; IQR = −0.88, 0.40) versus without SWDs (median = 0.42; IQR = −0.12, 0.70; *p*-value = 0.04). Statistically significant correlations were also found between worse sleep on multiple SDSC subscale scores and worse outcomes (Appendix A). Specifically, moderate correlation was found between worse (higher) GEC and Disorders of Initiation and Maintenance of Sleep (rs = 0.48), Sleep Wake Transition Disorders (rs = 0.33), Disorders of Excessive Somnolence (rs = 0.47), and worse total SDSC score (rs = 0.52). Moderate correlation was also found between worse (lower) NCI scores and Disorders of Excessive Somnolence (rs = −0.31), Sleep Hyperhidrosis (rs = −0.34), and worse total SDSC score (rs = −0.33). Weak to no significant correlation was found between TBI severity markers (ISS, GCS, head AIS) and SDSC domain scores, the GEC, or the NCI (Appendix A).

Among the 100 patients with complete GEC data, 25 (25%) had clinically relevant t-scores ≥65, indicating parent-observed behaviors consistent with executive dysfunction, and 88% of these patients had SWDs. In bivariate analyses (Table 3), pre-admission chronic comorbidities and presence of SWDs at follow-up were significantly associated with worse GEC (both *p* < 0.05). When controlling for ISS, chronic comorbidity, and WRAT-4/5 word reading score, the presence of a SWD was associated with significantly worse GEC (β = 7.76; 95% CI = 2.47, 13.06; Table 4). Pre-admission neurodevelopmental and psychiatric conditions also remained significant in the final model.

Among 79 patients with complete data for the NCI, 18% had scores ≤−1.0, and 86% of those patients had SWD. In bivariate analyses (Table 3), Medicaid insurance, higher ISS, pre-admission chronic comorbidities, worse WRAT-4/5 word reading scores, frequent headaches, and SWD at follow-up were significantly associated with worse NCI scores (all *p* < 0.05). When controlling for covariates, the presence of SWD was significantly associated with worse NCI scores (β = −0.4; 95% CI = −0.76, −0.04; Table 5). Weekly or more frequent headaches, WRAT-4/5 word reading score, and ISS outcomes were also significantly associated with the NCI in the final model.

## 4. Discussion

Children surviving critical care for TBI have high rates of SWD and are vulnerable to long-term executive dysfunction. The present study contributes to the pediatric critical care and TBI literature by providing novel information regarding the relationship between parent-reported sleep outcomes and executive functioning measured through subjective parent report (GEC) and through objective neuropsychological assessment (NCI) in a cohort of pediatric TBI survivors during the acute recovery phase after critical care. Overall, and in support of the study hypothesis, SWD in children with TBI was associated with significantly worse executive function outcomes after hospital discharge; as such, SWDs may serve as modifiable targets to improve outcomes.

Related to prevalence, SWD was present in 68% (*n* = 89) of our clinical sample using a sleep measure validated in pediatric patients with brain injury following critical care—the SDSC [43]. This figure represents an increase from our previous pilot work using the SDSC, which showed that 52% of youth with TBI presenting to our PICU follow-up clinic have SWDs one to three months post-hospital discharge [5]. This also represents an increase in prevalence reported in orthopedic injury controls and the approximately 40% of healthy pediatric cohorts reported to have SWD [5,12,13]. Other works in the pediatric TBI population report SWDs develop in at least 20% of pediatric TBI patients within the first six months of hospital discharge, though variability in measurement tools may account for the majority of the differences between studies, particularly as most prior studies did not use a validated sleep measure like the SDSC [12]. The current study showed that 53% of the entire cohort exhibited phenotypes consistent with insomnia and circadian disturbance, which is remarkably congruent with our previous work [5], as well as the work of others, highlighting initiation and maintenance of sleep as persistent issues in youth after TBI of all severities [12].

Our results also showed no significant differences between TBI severity groups and SWDs when assessed overall or with SDSC domains, consistent with prior works that used validated sleep tools [12]. We also showed that youth exhibit important SWDs in multiple domains regardless of their TBI severity. While our study did not measure baseline sleep prior to the injury, our results are consistent with prior works, suggesting the TBI results in the worsening of pre-existing SWDs or development of new SWDs given the remarkably high prevalence. This observation supports the need for increased evaluation and intervention for sleep following TBI. However, our cross-sectional analysis was unable to determine causation. Children following PICU admission are at risk of a variety of other morbidities, such as pain and psychological trauma, which could additionally contribute to the development of SWDs, and the mediating effects of these outcomes should be evaluated in future studies [32].

The risk of TBI related sequela within the executive function system is an important area of consideration when working with youth during the acute phase of recovery. The term executive function is commonly used to describe complex neurocognitive processes such as a person’s ability to regulate attention and concentration, self-monitor, plan, organize, utilize cognitive flexibility, engage in abstract reasoning, problem-solve, inhibit impulses, initiate tasks, and regulate emotions [60]. Given the importance of the executive function system, it is clear why it is critical for academic achievement and general childhood development [6,7,8,9,10,11,14,15,16,17,18,19]. Unfortunately, the executive function system is vulnerable to dysfunction after a TBI. Prior work shows that executive function impairments are present during the acute recovery phase, and that problems persist long-term and are significantly worse in youth with complicated mild, moderate, or severe TBI who required critical care hospitalization in comparison to healthy children, those with concussion, and orthopedic injury controls [20,21,27,28,29,30,57,61,62]. Congruent with expectations grounded in the research literature, we found that 25% of children with TBI in our sample had clinically elevated GEC scores when executive function was measured by parent report. Additionally, we found 18% exhibited notable dysfunction on objective measures of executive functions using the NCI. Risk factors for impairment varied by outcome measure in our study, likely due to the difference in measurement techniques, as discussed below.

Prior research has listed risk factors for cognitive/executive function impairment to include markers of TBI severity [20,29,57,58]. Severity of TBI is frequently classified based on hospital admission GCS score, with scores of 13–15 reflecting mild TBI, 9–12 reflecting moderate TBI, and 3–8 reflecting severe TBI [33], but is ideally classified using multiple indicators [63]. Mild TBI is further categorized as complicated mild TBI when abnormalities on imaging including fracture and hemorrhage are present compared to (uncomplicated) mild TBI, which is not associated with visible abnormalities on structural neuroimaging. Our study found weak associations between outcomes and TBI severity, although this varied with parent report versus objective measures and by severity measures (e.g., GCS versus ISS). GCS at admission is known to have multiple confounders, limiting its utility in predicting TBI outcomes, though the strongest associations are found with more severe injuries (GCS ≤8) [41,64,65]. Our study similarly showed trends for worsening outcomes among the patients classified as severe TBI based on GCS score, but no significant relationship between GCS and outcomes overall. Additional metrics of severity include ISS and AIS scores; these outcomes are objectively assigned ratings based on radiographic reports and administrative diagnoses, and some prior work suggests improved classification of illness severity based on outcomes using ISS versus GCS, especially for those with less severe TBI [66,67]. Prior work in hospitalized pediatric TBI patients showed worse outcomes among children with polytrauma versus isolated TBI, which is accounted for by the ISS [41,67]. Our study similarly found the strongest associations between severity and outcomes when using ISS, though the strength of the relationship varied by outcome measured, consistent with prior work [59,66,68]. We found a significant relationship between ISS quartiles and objective measures of executive function (NCI), but no relationship between ISS and parent report with the GEC.

Markers of TBI severity are not the only risk factors mentioned in the research literature related to the paths of recovery. In fact, cognitive recovery trajectories after pediatric TBI are variable and often individualized. Known risk factors for deficits in executive dysfunction beyond severity include age at injury, location of the injury in the brain, socioeconomic status, family functioning, and pre-injury health and neurodevelopmental conditions [21,24,57,58,62,69,70]. Few, if any, of the known risk factors for long-term neurocognitive dysfunction secondary to TBI are modifiable with intervention. Additionally, children surviving critical care hospitalization are exposed to multiple risk factors that may compound cognitive morbidity beyond initial TBI severity, including but not limited to neuroactive sedative and analgesic medications, mechanical ventilation, hypoxia, systemic inflammation, and seizures [71,72]. Importantly, our study identified potentially modifiable risk factors for worse outcomes with SWDs. Sleep/wake disturbances were strongly associated with dysfunction on the GEC when controlling for covariates. Sleep/wake disturbances and frequent headaches were significantly associated with worse outcomes when controlling for other covariates on the NCI. There is a known bidirectional relationship between SWDs and headaches [73,74,75,76], although our study results indicate that both contribute to worse executive function outcomes measured by the NCI. Sleep is potentially modifiable, and treatment of SWDs are often used in the management of headaches [73,77]. Targeted evaluation and intervention for SWDs early in the recovery process may be particularly important for children with TBI in that good restorative sleep promotes neuronal healing and reduces inflammation, which could maximize functional recovery [4].

More specifically, in our study, poorer outcomes were associated with specific domains of SWD. On the GEC, the most notable associations were within sleep domains related to insomnia, circadian disturbance, and excessive daytime sleepiness, while objective executive dysfunction on the NCI was most notably related to excessive daytime sleepiness. Prior work in pediatric TBI patients shows similar associations between excessive daytime sleepiness and worse cognitive outcomes [17,78]. All of the aforementioned sleep domains are ripe for targeting with evidenced based interventions, with the goal of not only promoting better sleep and daytime functioning, but perhaps improving cognitive processes such as executive function within this vulnerable patient population. Unfortunately, a recent systematic review showed no studies evaluating interventions to improve sleep after TBI hospitalization in children [12]. A recent randomized clinical trial of outpatient youth with concussion studying melatonin found no improvement in post-concussive symptoms overall, but did show improvements in some sleep outcomes, though this trial did not include children with pre-injury comorbidities or more complicated injuries [79,80]. The findings of our current study advocate for including sleep assessment in standard TBI care, and working to identify effective interventions for SWDs.

As stated, we noted variability in our results based on how executive function was measured, using the GEC (parent-report) or the NCI (objective assessment). We assert that both outcomes have merit and add useful information to research and clinical care, but are measuring different outcomes related to executive functions, which accounts for the differences we noted [81,82]. Neuropsychological evaluation relies on measurements of direct individual performance using a variety of standardized assessment tools, such as those included in the NCI of our study, to quantify cognitive/executive functioning capabilities. Neuropsychologists also use parent/caregiver report of behaviors conceptually related to executive function, such as those reported in the BRIEF using the GEC. Both the objective measures and subjective reports are traditionally used in combination by neuropsychologists to understand the impact of the TBI on cognitive skills and manifestations of those skills in daily functioning so that appropriate accommodations and interventions can be developed and deployed. The NCI outcome in our study represents a robust individually administrated objective measure of overall executive functioning performance, combining the aforementioned individually administered objective assessment tools, consistent with prior work [32]. The risk factors for worse NCI in our study were associated with objective injury severity (ISS) and measures of baseline cognitive performance, as would be expected based on prior TBI literature. The authors of the BRIEF-2 broadly define executive functioning as a skill set that involves concurrent modulation of behavioral, emotional, and metacognitive skills, which can be observed in daily functioning to describe the day-to-day components and presentations of executive functioning through proxy report [44]. As such, the GEC in our study represents the aforementioned parent/caregiver report of behavioral aspects of executive functioning, rather than objective assessment. Indeed, we found different results using the GEC versus the NCI; the GEC results showed no association with objective injury severity or baseline cognitive function, but strong associations with pre-injury comorbidity and SWDs. The differences between results for our different outcome measures highlight the need for researchers to carefully consider measurement tools when designing studies and interpreting the results for cognitive outcomes after TBI.

## 5. Limitations

Our study has several limitations to consider. It is a single-institution, retrospective cohort study with a relatively small sample size due to the unique population of interest. Heterogeneity in pediatric TBI populations is unavoidable, and there are known geographical and institutional differences in populations and acute treatments in pediatric critical care and TBI populations, which may limit generalizability of our findings [2,83,84]. Additionally, The SDSC, the BRIEF-P, and the BRIEF-2 are parent-completed measures. Research demonstrates relying on third party ratings for a child’s behavior increases the likelihood of response bias due to parent perceptions, which are influenced by parent demographic backgrounds and societal norms [85,86]. As such, the outcomes that parent-completed measures generate can be influenced by many factors, including those not directly related to each measure’s intended construct—thus making it challenging to capture all of the variance within multivariable models. Further, our study was a cross-sectional analysis limiting evaluation of causation between exposures and outcomes. We assessed parent-reported sleep and executive functioning, as well as direct assessment of executive functioning at one to three months post-hospital discharge; longitudinal studies are needed to assess how the observed relationships change over time. It is also important to note individual impact of factors outside the scope of clinical study (i.e., socioeconomic status and stress related to the COVID-19 pandemic) may have influenced the results of this study [87,88].

## 6. Conclusions

Youth with TBI requiring critical care hospitalization have high rates of SWDs that are associated with worse cognitive outcomes, measuring aspects of executive function. Sleep/wake disturbances were strongly associated with both a behaviorally based outcome measure of executive function (GEC) and when utilizing objective measures of executive function (NCI) during neuropsychological assessment. Our study highlights the need for increased screening and intervention for SWDs in the acute recovery phase following TBI hospitalization, and the need to understand how SWDs and sleep interventions can impact cognitive recovery trajectories after pediatric TBI.

## Figures and Tables

**Table 1 children-09-00748-t001:** Demographic and clinical characteristics of the overall cohort and by sleep disturbance group.

	All	Sleep Disturbance	Normal Sleep	X^2^ or U	*p*-Value
	N = 131 (%)	*n* = 89 (%)	*n* = 42 (%)		
Age, Median years (IQR)	11.5 (7.4, 13.8)	11.5 (7.2, 14.3)	11.0 (7.7, 13.7)	1881	0.95
Male sex	78 (60%)	58 (65%)	20 (48%)	3.65	0.06
Race				9.00	0.17
White	94 (72%)	62 (70%)	32 (76%)		
Asian	5 (4%)	4 (5%)	1 (2%)		
Pacific Islander	3 (2%)	1 (1%)	2 (5%)		
African American	1 (1%)	0 (0%)	1 (2%)		
American Indian or Alaska Native	1 (1%)	1 (1%)	0 (0%)		
More than one race	9 (7%)	9 (10%)	0 (0%)		
Declined or not reported	18 (14%)	12 (14%)	6 (14%)		
Hispanic ethnicity	13 (10%)	6 (7%)	7 (17%)	3.15	0.11
Medicaid insurance	79 (60%)	52 (58%)	27 (64%)	0.41	0.52
Pre-injury chronic condition					
Medical	12 (9%)	8 (9%)	4 (10%)	0.01	>0.99
Psychiatric	13 (10%)	10 (11%)	3 (7%)	0.54	0.55
Neurodevelopmental	23 (18%)	21 (24%)	2 (5%)	6.99	0.01
Critical care intervention, any	61 (47%)	42 (47%)	19 (45%)	0.04	0.83
Mechanical ventilation	39 (30%)	25 (28%)	14 (33%)	0.38	0.54
Arterial line	11 (8%)	4 (4%)	7 (17%)	5.5	0.04
Central venous line	8 (6%)	4 (5%)	4 (10%)	1.26	0.27
ICP monitor	7 (5%)	2 (2%)	5 (12%)	5.26	0.03
Neurosurgical intervention	11 (8%)	7 (8%)	4 (10%)	0.1	0.74
Other surgical intervention	37 (28%)	24 (27%)	13 (31%)	0.22	0.64
Hyperosmolar therapy	4 (3%)	3 (3%)	1 (2%)	0.09	>0.99
Targeted temperature management	3 (2%)	0 (0%)	3 (7%)	6.51	0.03
Antiepileptic infusion	9 (7%)	6 (7%)	3 (7%)	0.01	>0.99
Hemodynamic resuscitation	6 (5%)	3 (3%)	3 (7%)	0.93	0.39
Median hours mechanical ventilation (IQR)	4.5 (2.6, 25.1)	4.2 (2.8, 24.0)	6.2 (2.3, 203.1)	146	0.39
Any seizure	18 (14%)	11 (12%)	7 (17%)	0.45	0.59
Length of stay, Median days (IQR)					
Critical care	1.0 (0.7, 2.1)	1.0 (0.6, 2.2)	1.4 (0.7, 1.9)	547	0.77
Hospital	2.6 (0.9, 5.5)	2.8 (1.1, 5.4)	1.7 (0.9, 5.8)	1931	0.76
Inpatient rehabilitation discharge	9 (7%)	4 (5%)	5 (12%)	2.45	0.15
Median days from discharge to follow up (IQR)	49 (38, 69)	49 (38, 67)	49 (38, 74)	1838	0.88
Functional Status Scale, Median (IQR)					
Pre-admission baseline	6 (6, 6)	6 (6, 6)	6 (6, 6)	1869	0.49
Follow-up	6 (6, 6)	6 (6, 6)	6 (6, 7)	1713	0.42
Worsening in Functional Status Scale	25 (19%)	15 (17%)	10 (24%)	0.97	0.35
Glasgow Coma Scale, Median (IQR)	15 (14, 15)	15 (14, 15)	15 (12, 15)	2056	0.29
Brain injury severity category by GCS				4.28	0.23
Mild (13–15)	56 (43%)	36 (40%)	20 (48%)		
Mild Complicated (13–15, radiographic injury)	49 (37%)	38 (43%)	11 (26%)		
Moderate (9–12)	12 (9%)	6 (7%)	6 (14%)		
Severe (3–8)	14 (11%)	9 (10%)	5 (12%)		
Injury Severity Scale, Median (IQR)	11 (6, 19)	14 (6, 21)	10 (5, 17)	2041	0.15
Abbreviated Injury Scale ^a^, Median (IQR)					
Head/neck, *n* = 101	3 (2, 4)	3 (2, 4)	3 (1, 4)	1251	0.48
Face, *n* = 29	2 (1, 2)	2 (1, 2)	1 (1, 2)	97	0.33
Chest, *n* = 28	3 (2, 3)	3 (2, 3)	3 (2, 3)	77	0.88
Abdomen/pelvis, *n* = 22	3 (2, 4)	3 (2, 4)	2 (2, 3)	41	0.71
Extremity, *n* = 37	2 (2, 3)	2 (2, 2)	3 (2, 3)	88	0.05
External, *n* = 115	1 (1, 1)	1 (1, 1)	1 (1, 1)	1455	0.58
Mechanism of injury				5.01	0.66
Fall	36 (28%)	22 (25%)	14 (33%)		
Motor vehicle accident	34 (26%)	26 (29%)	8 (19%)		
Auto-pedestrian/bike	16 (12%)	11 (12%)	5 (12%)		
Bicycle, skateboard, scooter	21 (16%)	14 (16%)	7 (17%)		
ATV	9 (7%)	6 (7%)	3 (7%)		
Other blunt	12 (9%)	8 (9%)	4 (10%)		
Penetrating	2 (2%)	2 (2%)	0 (0%)		
Unknown	1 (1%)	0 (0%)	1 (2%)		
Type of intracranial injury on imaging					
No hemorrhage or fracture	56 (43%)	38 (43%)	18 (43%)	0.001	>0.99
Subdural	26 (20%)	18 (20%)	8 (19%)	0.03	>0.99
Subarachnoid	21 (16%)	15 (17%)	6 (14%)	0.14	0.8
Epidural	11 (8%)	10 (11%)	1 (2%)	2.91	0.1
Contusion	25 (19%)	14 (16%)	11 (26%)	2.02	0.16
Diffuse axonal injury	11 (8%)	6 (7%)	5 (12%)	0.99	0.33
Mixed or multiple in same location	11 (8%)	4 (5%)	7 (17%)	5.5	0.04
Indeterminate	5 (4%)	4 (5%)	1 (2%)	0.35	>0.99
Location of intracranial injury on imaging					
Frontal	37 (28%)	25 (28%)	12 (29%)	0.003	0.95
Parietal	21 (16%)	15 (17%)	6 (14%)	0.14	0.8
Temporal	26 (20%)	14 (16%)	12 (29%)	2.96	0.09
Occipital	20 (15%)	14 (16%)	6 (14%)	0.05	>0.99
Cerebellum	3 (2%)	2 (2%)	1 (2%)	0.002	>0.99
Abusive injury suspected/confirmed	3 (2%)	1 (1%)	2 (5%)	1.69	0.24
Headaches weekly or more frequent at follow-up	22 (17%)	17 (19%)	5 (12%)	1.06	0.45
New psychological diagnosis at follow-up	32 (24%)	26 (29%)	6 (14%)	3.44	0.08

Sleep disturbance groups were compared with Chi-square tests with Fischer Exact correction for expected counts <10 for categorical variables, and with Mann–Whitney U tests for continuous variables. ^a^: Abbreviated Injury Scale distributions for patients with injuries in that area (body region score > 0); X^2^: Chi-square statistic; U: U-statistic result of Mann–Whitney U comparisons; ATV: all-terrain vehicle; GCS: Glasgow Coma Scale; IPC: intracranial pressure; IQR: interquartile range.

**Table 2 children-09-00748-t002:** Average scores on cognitive outcomes by sleep disturbance group.

	All Mean (SD)	Normal Sleep Mean (SD)	Sleep Disturbance Mean (SD)	t(df)	*p*-Value
GEC T-score, *n* = 100	53.89 (12.43)	47.42 (9.48)	56.8 (12.55)	−3.7 (98)	<0.001
Neurocognitive Index (NCI) z-score, *n* = 79	−0.05 (1.01)	0.24 (0.91)	−0.20 (1.04)	1.9 (77)	0.06
Numbers combined, *n* = 108	8.40 (2.73)	8.78 (2.67)	8.20 (2.76)	1.1 (106)	0.29
Lists Immediate, *n* = 110	8.50 (2.70)	8.67 (2.94)	8.42 (2.59)	0.5 (108)	0.67
Lists Delayed, *n* = 111	8.60 (3.29)	8.76 (3.48)	8.53 (3.22)	0.3 (109)	0.74
DKEFS number letter switching, *n* = 85	7.21 (4.05)	8.10 (3.32)	6.75 (4.33)	1.5 (83)	0.11
DKEFS category fluency, *n* = 88	9.69 (3.53)	10.06 (3.90)	9.49 (3.33)	0.7 (86)	0.49
DKEFS letter fluency, *n* = 87	8.14 (2.81)	7.90 (2.66)	8.27 (2.91)	−0.6 (85)	0.56
Combined coding, *n* = 105	8.31 (3.09)	8.97 (2.96)	8.00 (3.13)	1.5 (103)	0.13
Combined symbol search, *n* = 105	9.59 (3.39)	10.06 (2.44)	9.37 (3.75)	1.0 (103)	0.26
Word reading, *n* = 111	97.97 (16.02)	99.76 (17.38)	97.09 (15.36)	0.8 (110)	0.43

Sleep disturbance groups were compared with independent *t*-tests using 2-sided *p*-value; t(df): t-statistic (degrees of freedom); GEC: Global Executive Composite; DKEFS: Delis Kaplan Executive Function System.

**Table 3 children-09-00748-t003:** Association between cognitive outcomes and demographic and clinical characteristics using simple linear regression for each outcome.

	Global Executive Composite, *n* = 100 β-Coefficient (95% Confidence Interval)	Neurocognitive Index, *n* = 79 β-Coefficient (95% Confidence Interval)
Age in years	0.42 (−0.17, 1.00)	−0.03 (−0.11, 0.05)
Male sex	3.21 (−1.74, 8.17)	−0.29 (−0.75, 0.17)
White race	2.89 (−2.49, 8.27)	−0.24 (−0.77, 0.31
Hispanic ethnicity	−5.62 (−14.21, 2.97)	0.23 (−0.63, 1.09)
Medicaid insurance	1.93 (−3.06, 6.92)	−0.76 (−1.19, −0.33) *
Pre-injury chronic condition, any	6.85 (1.44, 12.26) *	−0.53 (−1.00, −0.07) *
Medical	−3.79 (−12.42, 4.84)	−0.46 (−1.17, 0.25)
Psychiatric	11.12 (3.17, 19.08) *	−0.52 (−1.20, 0.15)
Neurodevelopmental	13.38 (7.17, 19.58) *	−0.81 (−1.32, −0.30) *
Word reading (baseline estimate, log score)	−23.2 (−66.61, 20.21	8.27 (5.81, 10.73) *
Critical care intervention, any	1.97 (−2.98, 6.93)	0.04 (−0.42, 0.49)
Mechanical ventilation	3.80 (−1.61, 9.21)	0.17 (−0.31, 0.65)
Any seizure	0.93 (−8.21, 10.07)	−0.11 (−0.68, 0.46)
Length of stay (LN days)		
Critical care	0.61 (−3.14, 4.36)	0.20 (−0.15, 0.54)
Hospital	1.07 (−1.14, 3.29)	−0.18 (−0.39, 0.03)
Inpatient rehabilitation discharge	2.60 (−7.83, 13.03)	−0.33 (−1.19, 0.53)
Time from discharge to follow-up, days	−0.03 (−0.14, 0.08)	0.001 (−0.01, 0.01)
Functional Status Scale, any worsening	−0.19 (−6.43, 6.06)	−0.07 (−0.63, 0.49)
Injury severity by Glasgow Coma Scale		
Mild (13–15)	Reference	Reference
Mild complicated (13–15, +imaging)	0.04 (−5.46, 5.55)	−0.01 (−0.54, 0.52)
Moderate (9–12)	0.23 (−9.23, 9.69)	−0.33 (−1.23, 0.57)
Severe (3–8)	6.51 (−2.49, 15.51)	−0.16 (−0.87, 0.54)
Injury Severity Scale scores		
Lowest quartile	Reference	Reference
Middle quartiles	3.25 (−4.42, 10.91)	−0.74 (−1.33, −0.15) *
Highest quartile	7.05 (−1.59, 15.67)	−0.73 (−1.40, −0.05) *
Abbreviated Injury Scale (0–2 reference)		
Head ≥ 3–5	2.44 (−2.55, 7.42)	−0.25 (−0.70, 0.21)
Chest ≥ 3–5	1.37 (−5.77, 8.51)	−0.34 (−1.00, 0.31)
Abdomen pelvis ≥ 3–5	4.41 (−3.46, 12.29)	0.01 (−0.74, 0.77)
Extremity ≥ 3–5	−1.91 (−9.27, 5.45)	−0.16 (−0.82, 0.50)
Mechanism of injury		
Fall	Reference	Reference
Motor vehicle accident	6.94 (−0.01, 13.88)	0.11 (−0.53, 0.75)
Auto-pedestrian/bike	7.49 (−1.22, 16.21)	−0.48 (−1.29, 0.34)
Bicycle, skateboard, scooter	1.41 (−6.41, 9.24)	0.52 (−0.19, 1.24)
All terrain vehicle	0.16 (−9.46, 9.78)	0.03 (−0.82, 0.87)
Other	5.74 (−2.97, 14.46)	0.61 (−0.21, 1.42)
Abusive injury suspected or confirmed	5.21 (−1.98, 12.40)	0.40 (−0.61, 1.42)
Headaches more than weekly frequency	0.79 (−5.53, 7.10)	−0.81 (−1.32, −0.30) *
New psychological diagnosis	3.38 (−2.10, 8.86)	−0.20 (−0.76, 0.35)
Sleep disturbance present	9.38 (4.36, 14.40) *	−0.44 (−0.91, −0.03) *

* *p* < 0.05.

**Table 4 children-09-00748-t004:** Multiple linear regression analysis of Global Executive Composite outcome.

KERRYPNX	Full Model Beta Coefficient (95% Confidence Interval)	Reduced Model Beta Coefficient (95% Confidence Interval)
Injury Severity Scale		--
Lowest quartile	Reference	
Middle quartiles	2.97 (−3.94, 9.88)	
Highest quartile	3.65 (−4.22, 11.52)	
Word reading (log score)	−7.60 (−51.17, 35.97)	--
Pre-injury Psychiatric Condition *	8.41 (−0.20, 17.01)	9.02 (0.88, 17.15)
Pre-injury Neurodevelopmental Condition *	8.04 (0.76, 15.31)	8.58 (1.68, 15.48)
Sleep disturbance present *	8.04 (2.49, 13.59)	7.76 (2.47, 13.06)

Backward stepwise regression used for reduced model at *p* > 0.1 threshold for removal; Final reduced model statistics: F = 11.04, *p*-value ≤ 0.001; R2 = 0.30; adjusted R2 = 0.28; * *p* < 0.05 in final model.

**Table 5 children-09-00748-t005:** Multiple linear regression analysis of Neurocognitive Index outcome.

	Full Model Beta Coefficient (95% Confidence Interval)	Reduced Model Beta Coefficient (95% Confidence Interval)
Medicaid insurance	−0.24 (−0.61, 0.13)	−−
Pre-injury neurodevelopmental condition	−0.30 (−0.73, 0.13)	−−
Word reading (log score) *	5.73 (3.21, 8.25)	6.37 (3.89, 8.85)
Injury Severity Scale score		
Lowest quartile	Reference	Reference
Middle quartiles *	−0.48 (−0.95, −0.004)	−0.59 (−1.05, −0.13)
Highest quartile *	−0.48 (−1.03, 0.07)	−0.60 (−1.12, −0.07)
Headache weekly or more frequent *	−0.48 (−0.88, −0.07)	−0.53 (−0.93, −0.12)
Sleep disturbance present *	−0.32 (−0.69, 0.05)	−0.40 (−0.76, −0.04)

Backward stepwise regression used for reduced model at *p* > 0.1 threshold for removal; Final reduced model statistics: F = 12.7, *p*-value ≤ 0.001; R2 = 0.47; adjusted R2 = 0.44; * *p* < 0.05 in final model.

## Data Availability

The data presented in this study are available on request from the corresponding author. The data are not publicly available due to protected health information, but de-identified data can be made available on request.

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
