# Peer review of "Sleep and Executive Functioning in Pediatric Traumatic Brain Injury Survivors after Critical Care"

_children, 2022, doi:10.3390/children9050748_

Round 1
Reviewer 1 Report
children-1722811
This study explored relationship between youth traumatic brain injury (TBI) and sleep-wake disorders (SWD). TBI severity and other factors were revealed as risk factors for SWD. This research provides a good background data for outcome of children’s TBI. Please find my comments below.
1) In terms of description, this study is very comprehensive and informative. Potential causes of the SWD, however, was not explored enough. Is it possible to infer whether the injury itself caused SWD or psychological impact of the trauma caused SWD?
2) The results seem to lack objective measures such as brain imaging and blood tests. Providing them would strengthen the conclusions.
Author Response
Reviewer 1:
- This study explored relationship between youth traumatic brain injury (TBI) and sleep-wake disorders (SWD). TBI severity and other factors were revealed as risk factors for SWD. This research provides a good background data for outcome of children’s TBI.
-
- Thank you for the encouragement.
- In terms of description, this study is very comprehensive and informative. Potential causes of the SWD, however, was not explored enough. Is it possible to infer whether the injury itself caused SWD or psychological impact of the trauma caused SWD?
-
- Given the cross sectional design of this study’s analysis, we cannot infer causation between TBI and the observed SWDs. Prior studies have shown that TBI leads to SWDs in adults and children, and that SWDs are not fully explained by psychological impacts alone. We attempted to explore this issue by assessing differences in SWDs by pre-existing psychological or neurodevelopmental diagnoses, as well as new psychological diagnoses made at follow-up by our expert clinical team. Interestingly, while pre-existing neurodevelopmental conditions did increase the risk for SWDs, the development of new psychological diagnoses did not show a statistically significant difference between SWD groups (Table 1). We have added additional data to the text in the results section to highlight this point: “No significant difference was found for development of new psychological diagnoses (e.g. depression, anxiety, stress disorders) between SWD groups.”
- The results seem to lack objective measures such as brain imaging and blood tests. Providing them would strengthen the conclusions.
- We have added additional data to Table 1 regarding imaging findings in our cohort. Unfortunately, we did not collect biomarker data from blood tests as part of this
Reviewer 2 Report
The authors describe an important topic in child TBI care, which was neglected for a long time. Their current contribution is in line with earlier presented studies.
The manuscript is mainly well written. I have some minor comments and one more substansial comment.
In lines 88-90, the authors describe the scales used to identify the severity of the TBI. I assume, they used the children versions of the scales (if available), although they didn't mention it. Please clarify.
The reference to the AIS (line 90) is rather odd: to the original publication out of 1971 (incorrect presented in the reference list), while the AIS has been updated several times since (see: https://www.aaam.org/abbreviated-injury-scale-ais/) . What version of the AIS has been used? Please, make the right reference.
In lines 104-123, the SDSC is presented. In the middle of this paragraph (lines 117-118), a single sentence refers to a validation study in a TBI population. I suggest to replace this sentence to line 106, immediatley after the introduction of the scale, so the reader knows immediately that this scale is appropriate for this population. The remark in line 288-289 about the validation study in young TBI patients can be removed. Furthermore, the remark about the internal consistency (line 118-119) should be removed to the results section.
The results section is rather difficult to read, because of the huge amount of data. So, I was hoping for more clear descriptions of the results and conclusions in the discussion section. However, I was rather disappointed, because the authors did choose to mention the instruments they used instead of the behaviours/functions these instruments should represent (see for instance lines 358 and 362). Moreover, they mention several times that the outcomes varied by outcome measures and severity measures, without clarifying the differences.
I suggest to rewrite important parts of the discussion, being more specific about the results (naming functions and behaviours instead of instruments) and describing the variations in outcomes more specific.
A thorough check of the reference list will reveal some minor mistakes (see i.e. 32 and 41). There will be more, I think.
Author Response
Reviewer 2:
- The authors describe an important topic in child TBI care, which was neglected for a long time. Their current contribution is in line with earlier presented studies. The manuscript is mainly well written. I have some minor comments and one more substantial comment.
-
- Thank you for the encouragement. We have addressed your comments below.
- In lines 88-90, the authors describe the scales used to identify the severity of the TBI. I assume, they used the children versions of the scales (if available), although they didn't mention it. Please clarify.
-
- Thank you for pointing out our omission here. Yes, we used the pediatric version of the Glasgow Coma Scale and have clarified this in the methods.
- The reference to the AIS (line 90) is rather odd: to the original publication out of 1971 (incorrect presented in the reference list), while the AIS has been updated several times since (see: https://www.aaam.org/abbreviated-injury-scale-ais/). What version of the AIS has been used? Please, make the right reference.
-
- We have clarified our trauma program utilizes the updated version of AIS (2015) for the patients described in this study. We have added additional references regarding accessing the 2015 version of AIS and National Trauma Data Bank standards used at our institution, consistent with standard practice for pediatric Level 1 trauma centers in the US. We have retained the original reference to the AIS and ISS development as is standard when citing these measures.
- In lines 104-123, the SDSC is presented. In the middle of this paragraph (lines 117-118), a single sentence refers to a validation study in a TBI population. I suggest to replace this sentence to line 106, immediately after the introduction of the scale, so the reader knows immediately that this scale is appropriate for this population. The remark in line 288-289 about the validation study in young TBI patients can be removed. Furthermore, the remark about the internal consistency (line 118-119) should be removed to the results section.
-
- Thank you for this suggestion. We have moved the validation sentence to the beginning of the paragraph within the methods. We have retained the sentence about validation in the discussion as it is germane to the point that prior works often do not use validated sleep measures in this population, which is an important consideration when comparing to prior literature. We have moved the internal consistency sentence to the results.
- The results section is rather difficult to read, because of the huge amount of data. So, I was hoping for more clear descriptions of the results and conclusions in the discussion section. However, I was rather disappointed, because the authors did choose to mention the instruments they used instead of the behaviours/functions these instruments should represent (see for instance lines 358 and 362). Moreover, they mention several times that the outcomes varied by outcome measures and severity measures, without clarifying the differences.
-
- We have added clarity throughout the discussion as to the different results based on different measures. We use the discussion to more fully explain the measurements/instruments in this study to provide context for why results are different between parent report and objective assessments of cognition, and for severity measures based on objective tools.
- I suggest to rewrite important parts of the discussion, being more specific about the results (naming functions and behaviours instead of instruments) and describing the variations in outcomes more specific.
-
- Thank you for the suggestion, we have added clarity throughout the discussion.
- A thorough check of the reference list will reveal some minor mistakes (see i.e. 32 and 41). There will be more, I think.
-
- Thank you for this review. We have corrected the formatting issues.
Reviewer 3 Report
In the present study, the Authors' primary aim was to evaluate the relationships between parent reported outcomes related to sleep and executive functioning in a cohort of pediatric traumatic brain injury (TBI) survivors during the acute recovery phase after critical care. Besides, the secondary aim was to explore the relationship between parent reported sleep outcomes and direct clinical assessment of executive functioning utilizing neuropsychological evaluations.
Overall, I found this study timely, original, well conducted and scientifically sound. I have some suggestions aimed to improve the quality of the paper and these are outlined below:
1) In the introduction a very brief note on the possible effects of TBI and onset of severe psychiatric disorders should be added with appropriate reference (see doi: 10.3390/brainsci11020275).
2) Were the participants consecutive or randomly selected? As well, how many subjects were screened, but refused to participate?
3) There aren't informations on exclusion criteria. Please, add this part or specify.
4) I believe that in Tables the report of only "p" value isn't scientifically sound. Please, report the full statistics.
Author Response
Reviewer 3:
- Overall, I found this study timely, original, well conducted and scientifically sound. I have some suggestions aimed to improve the quality of the paper and these are outlined below:
- Thank you for the encouragement.
- In the introduction a very brief note on the possible effects of TBI and onset of severe psychiatric disorders should be added with appropriate reference (see doi: 10.3390/brainsci11020275).
-
-
- Thank you for highlighting this issue. As noted in our response to Reviewer 1, we attempted to account for psychological disorders, but did not find a significant association with SWDs (Table 1). However, we note that our cross sectional analysis cannot assess causation, so have highlighted the need to consider other morbidities like psychological disorders in development of SWDs after TBI. We have added the following sentence to our discussion: “However, our cross sectional analysis was unable to determine causation. Children following PICU admission are at risk for a variety of other morbidities, such as pain and psychological trauma, that could additionally contribute to development of SWDs, and mediating effects of these outcomes should be evaluated in future studies.[32]”
-
- Were the participants consecutive or randomly selected? As well, how many subjects were screened, but refused to participate?
-
- We have added additional clarity regarding inclusion/exclusion to the methods section.
- There aren't informations on exclusion criteria. Please, add this part or specify.
-
- We have added additional clarity regarding inclusion/exclusion to the methods section.
- I believe that in Tables the report of only "p" value isn't scientifically sound. Please, report the full statistics.
-
- We have added the X2 or U values to Table 1. We have added the t statistic to Table 2. We have further defined these terms in table footnotes as well.
Round 2
Reviewer 2 Report
I thank the authors for their efforts to address all the comments.
I have no further comments. The manuscript can be published in this form.